# COVID-19 and Health Information Seeking Behavior: Digital Health Literacy Survey amongst University Students in Pakistan

**DOI:** 10.3390/ijerph18084009

**Published:** 2021-04-11

**Authors:** Rubeena Zakar, Sarosh Iqbal, Muhammad Zakria Zakar, Florian Fischer

**Affiliations:** 1Department of Public Health, University of the Punjab, Lahore 54590, Pakistan; rubeena499@gmail.com; 2Institute of Social and Cultural Studies, University of the Punjab, Lahore 54590, Pakistan; sarosh.iqbal@gmail.com; 3University of Okara, Okara 56300, Pakistan; mzzakir@yahoo.com; 4Institute of Public Health, Charité–Universitätsmedizin Berlin, 10117 Berlin, Germany; 5Institute of Gerontological Health Services and Nursing Research, Ravensburg-Weingarten University of Applied Sciences, 88250 Weingarten, Germany

**Keywords:** eHealth literacy, digital health literacy, sense of coherence, COVID-19, COVID-HL-Q, Pakistan

## Abstract

Amid the COVID-19 pandemic, digital health literacy (DHL) has become a significant public health concern. This research aims to assess information seeking behavior, as well as the ability to find relevant information and deal with DHL among university students in Pakistan. An online-based cross-sectional survey, using a web-based interviewing technique, was conducted to collect data on DHL. Simple bivariate and multivariate linear regression was performed to assess the association of key characteristics with DHL. The results show a high DHL related to COVID-19 in 54.3% of students. Most of the Pakistani students demonstrated ~50% DHL in all dimensions, except for reliability. Multivariate findings showed that gender, sense of coherence and importance of information were found to be significantly associated with DHL. However, a negative association was observed with students′ satisfaction with information. This led to the conclusion that critical operational and navigations skills are essential to achieve COVID-19 DHL and cope with stress, particularly to promote both personal and community health. Focused interventions and strategies should be designed to enhance DHL amongst university students to combat the pandemic.

## 1. Introduction

The widespread outbreak of the Severe Acute Respiratory Syndrome Coronavirus (SARS-CoV-2) poses a perpetual menace to public health. The World Health Organization (WHO) designated it as novel coronavirus disease 2019 (COVID-19) [1] and constituted this pandemic as a public health emergency of international concern. The COVID-19 pandemic has devastating effects, with more than 100 million confirmed cases and more than 2.3 million deaths across the globe until mid-February 2021 [2]. Currently, Pakistan is the 30th most affected country across the world and the 7th highest hit country in the Asia region [3], with more than 560,000 confirmed cases of COVID-19, as of 13 February 2021 [4].

The trajectory of daily reported cases in Pakistan revealed a progression of confirmed cases, with steady, high and low dips. A rise in the number of reported cases was evident from May 2020 onwards, where the highest peak was observed in June 2020 with more than 6000 cases in a day, followed by a steady decrease between July and September 2020 [4]. Later on, the second wave of the pandemic erupted and the government of Pakistan announced a second spell of COVID-19 on 28 October 2020, with a gradual increase in daily cases. Most recently, the highest spike of COVID-19 cases was reported on 6 December 2020 with more than 3700 cases per day [4]. Once again in January 2021, the number of COVID-19 cases began to increase, which requires critical attention [4].

With the rapid progression of the COVID-19 pandemic, there is wake of an “infodemic”, defined as a global epidemic of misinformation and disinformation [5]. Ironically, there is a tsunami of information, grabbing the public’s attention in the form of experts’ opinions, muffed by a barrage of misguided theories, half-baked piece of advice, sketchy remedies and rumors on social media platforms and other outlets. Further, the evolving scientific knowledge and research sometimes also raise controversies or reversals in infection prevention recommendations, in a relatively short span of time, crowding out the accurate public health management and making the general public more anxious [5,6,7]. For instance, the use of a cloth mask (e.g., cotton or gauze) was not recommended initially by WHO, nonetheless, in the light of emerging evidence, the Center for Disease Control and Prevention (CDC) recognized the effectiveness of cloth masks in slowing down the spread of COVID-19 [5]. 

Since the emergence of the pandemic of COVID-19, the disease has become a constant challenge for public health reforms. In the absence of effective treatment and limited availability of vaccination, preventive measures against COVID-19 are critical to control. However, there is a continuous influx of COVID-19 resources on the internet and social media, including both information and misinformation, serving as a double-edged sword [8]. Further, this plethora of complex and discordant information increases even more the spread of fear and anxiety, engendering more confusion, chaos and panic among the public than the virus itself [9]; this increases rates of depression, affects mental health, causes poor quality of life and harms the well-being of individuals [10]. Given the context of the current health dilemma, the issue of health literacy is crucial to tackle the disease, particularly to ensure preparedness of the healthcare system and mitigate its effects on individual and societal health. 

Health literacy entails knowledge, competence and skills of an individual to attain, process, communicate and comprehend health information and services to promote and improve personal and community health through effective health decisions [11,12,13,14,15]. Digital Health Literacy (DHL), also phrased as eHealth literacy, is an extension of health literacy within the context of technology or electronic sources of information to understand and address any health problem [15,16]. Health literacy is a significant empowerment strategy, enabling individuals to seek necessary information, control their own health matters and take responsibility for their actions [15,17]. In recent years, health literacy has gained significant attention, owing to its association with social determinants of health. The WHO commission on Social Determinants of Health also recognized health literacy in determining health inequalities within low- and middle-income countries [18]. Moreover, the adequacy of health literacy facilitates improving personal care, creating an enabling environment, implementing health policies and achieving health outcomes, particularly in reducing health inequity and stress [10,17,19]. Nevertheless, low health literacy results in poor self-management and miserable health outcomes [20].

Health literacy is a broader concept, comprising three domains: functional literacy, interactive literacy, and critical literacy [21]. Functional literacy indicates the task-based knowledge and skills to acquire and act on health information regarding defined risks, recommended use of health services and adherence [13,21]. Interactive literacy describes more advanced cognitive and social skills and competence to extract, comprehend and differentiate between varied sources of health information, through higher level of interaction with experts/professionals [13,21]. Critical health literacy represents the most advanced cognitive and social skills, and competence to critically analyze health information from varied sources and apply this information to exert effective control in both personal and community health [13,21]. Broadly, this classification differentiates the varied literacy skills, which progressively extend from individual health to societal health, enabling autonomy, ensuring active engagement in wider actions and improving health outcomes.

Recognizing the complexity of health systems, health literacy and DHL in particular is known as an emerging field of inquiry, where no one is considered fully health literate and requires necessary support to understand and act upon health information. Specifically, some groups of the population are more prone to low health literacy, such as old age adults, migrants, ethnic or racial minorities, less educated people, and people with a low socio-economic and poor health status [22]. Here, the group of young university students (between 17 to 24 years and above) cannot be overlooked, who despite being keener to learn new things and enhance DHL skills are not fully equipped to critically analyze information and take appropriate decisions for personal and community health. Due to massive misinformation, the infodemic pertaining to COVID-19 highlighted the absence of health literacy proficiency. In turn, it increases stress, anxiety and risks of other morbidities, particularly among young people [13,23,24]. 

Overall, there is a paucity of evidence with reference to health literacy and COVID-19 in the local context of Pakistan. A large strand of literature on health literacy, particularly DHL, represents western countries [10,25,26,27,28]. Given this backdrop, this research is an attempt to provide a knowledge base and build an understanding regarding DHL of young students in the education sector of Pakistan. The objective of this research paper is to assess information-seeking behaviors among university students, along with their ability to find relevant information and deal with DHL, and the factors associated with DHL in Punjab, Pakistan. The primary aim of this research is not only hypothesis testing, but also to explore students’ DHL and behaviors related to COVID-19. It is written from the intellectual position that measures DHL, while it is worth taking all relevant factors. 

This research article is structured into six sections. Section 1 (Introduction) sets the context for present research, followed by research objective. Section 2 (Literature Review) presents a review of relevant studies, highlighting contributions and identifying gaps that this research would bridge. Section 3 (Materials and Methods) presents the detailed research methodology, including study design and setting, measures and their operationalization, data analysis approach and ethical considerations. Section 4 exhibits the results, supported by tables and figure. Section 5 includes the discussion which critically analyzes results in the international context, along with limitations and future directions. Lastly, Section 6 presents the conclusion of this research. 

## 2. Literature Review

Health literacy is mainly concerned in medical or healthcare settings. Several studies have shown that health literacy is a prerequisite for informed health decisions [15,29,30,31]. Previous research has demonstrated the relevancy of DHL for quality of life [32,33,34], highlighting that high levels of self-perceived knowledge or skills enhance opportunities for quality of life. However, a lack of such knowledge or skills lead to adverse outcomes. Hsu and colleagues determined that DHL has a positive association with health behaviors [32]. Furthermore, Neter and Brainin found a relationship of DHL with the ability to self-manage chronic illness and treatment [33]. Broucke and colleagues concluded from a multi-country Health Literacy Survey (HLS-19) that DHL of the general population (above 18 years) is associated with their use of digital resources [15].

On the other hand, limited or inadequate health literacy is associated with low socio-economic status and poor health outcomes [35,36]. Meppelink found that people with limited health literacy usually find it more difficult to comprehend health information [37]. Berkman and colleagues claimed that low health literacy results in more hospitalizations, lack of understanding when interpreting health messages and following treatment plans [36]. In case of older adults, low health literacy worsens health status, leading to increased mortality [36]. 

The concept of health literacy has evolved over the last decade. The first health literacy measure, namely the Rapid Estimate of Adult Literacy in Medicine (REALM), assessed patients’ functional literacy [38]. Nutbeam expanded the conceptualization and informed that health literacy not only entails the ability to read and write, but also the ability to extract information and critically analyze it [39]. Recognizing the relevance of digital information in modern society, Nutbeam revealed two distinctive concepts of health literacy, i.e., clinical risk and personal asset, which stimulate an understanding of health communication in both clinical and community settings [40]. Some researchers also explained that health-related knowledge [41], numeracy [42] and motivation to process health information [39] also contribute to health literacy. Later on, Sørensen and colleagues extended the concept, expanding the comprehensive dimension of health literacy to a variety of skills to function in modern healthcare systems [43].

During the last few years, multiple measures have been developed to assess health literacy [43,44]. Van der Vaart and colleagues conducted a study on patients’ DHL with rheumatic diseases and highlighted that six competences are required to comprehend DHL, including operational and navigation skills, information and evaluation skills, and interactivity skills (ability to add self-generated content and assuring privacy) [45]. Most previous studies have used the eHealth Literacy Scale [46], with an ability to measure either one or two concepts comprehensively, such as seeking and appraising online information [15,47,48,49]. Duong and colleagues validated and applied the HLS-EU tool to determine comprehensive health literacy for the general public in several Asian countries (Indonesia, Malaysia, Kazakhstan, Myanmar, Taiwan and Vietnam) [50]. Van der Vaart and Drossaert further applied a more recent instrument, the Digital Health Literacy Instrument (DHLI), to measure operational skills, navigation skills, information searching, evaluating reliability, determining relevance, adding self-generated content and protecting privacy [34]. 

Most of the literature on DHL relies on self-reporting, which is, thereby, a subjective measurement [33,51,52,53,54]. Ghaddar and colleagues used an online survey to examine adolescent health literacy and its relationship with the credible source of online-based health information [51]. Diviani and colleagues applied a mixed method approach to gain insight into the relationship between DHL and evaluation of online-based health information. They concluded that people with varied DHL showed disparities in evaluation ability [53].

Regarding global and regional levels of health literacy, Paascha-Orlow and colleagues conducted a systematic review of 85 studies and concluded that 26% people in the United States of America had low general health literacy [24]. Similarly, limited health literacy was found amongst 47% in Europe [25], 58.3% in Spain [25], 59% in Australia [26] and 50.1% in Germany [27]. Further, Raja et al. (2019) also explored the prevalence of limited health literacy amongst Southeast Asian countries [55], where limited HL was found to be associated with sociodemographic characteristics. Rababah and colleagues also found that health literacy is influenced by sociodemographic characteristics [56]. Past studies have also shown that low scores on sociodemographic measures, particularly education and income status, hinder the attainment of DHL [52,57,58]. These statistics depict the situation of developed countries, however, these also highlight that the status of health literacy in developing countries could be more worrisome due to underdeveloped health and education systems, as well as middle to poor socioeconomic status in the majority of the general population. A study conducted in Asia informed that 93% of students had insufficient health literacy [59]. Another study in Turkey showed that 29.3% of students had insufficient health literacy [28]. Similar results are evident from a study carried out among adolescents in Pakistan in 2018, which indicates that around 26% of respondents had adequate, while 21% had very limited health literacy [60].

Given the present scenario of the COVID-19 pandemic, Abel and McQueen highlighted that critical health literacy is indispensable to recognize risk factors and adopt preventive measures [61]. The general population has a major responsibility to search for relevant information and apply it to everyday life. However, it is seen that people with limited health literacy are more confused in assessing the trustworthiness of information related to a disease, due to the vast amount of online-based data available [27]. Particularly those who have suspected COVID-19 symptoms struggle and may be affected by mental illnesses. In this regard, Nguyen et al. examined the effect of health literacy on depression and quality of life in Vietnam. This study acknowledged that people with suspected COVID-19 symptoms had a higher rate of depression and lower health-related quality of life [10]. 

Regarding the DHL, the education sector plays a pivotal role, equipping the young generation with latest digital platforms of information and communication. Students are considered to be frequent users and a rich source of user-generated content (UGC) on digital platforms including social networks. Due to the influx of information at varied sources, extracting useful data remains a challenge. Recognizing this issue, Saura and colleagues explored how useful knowledge can be extracted and visualized from samples of readily available UGC. Results provided insights related to new data analysis, visualization techniques and innovative educational trends to understand user behaviors and their use of new technologies [62]. Further, Saura and colleagues also identified the millennial generation (born 1981–1993) as extensive and active users of digital technologies and tools of information, including internet and social networks [63]. Therefore, it is critical to understand the students′ online behavior and DHL. 

The issue of health literacy is critical amongst university students, who represent a significant proportion of the youth. University students are primarily the population who use digital technologies and web-based information the most [64]. Dadaczynski and colleagues carried out a study amongst university students aged ≥18 years in Germany, using an online-based survey, and investigated students′ DHL and web-based information-seeking behavior during early days of COVID-19 [65]. This research found that although DHL is well-developed amongst university students, a significant proportion of students still face difficulties with certain abilities to evaluate information. Similarly, Rosário and colleagues evaluated the association between DHL related to COVID-19 and online information-seeking behavior among Portuguese university students and found that students′ perceived DHL score was between good and very good [66]. Juvinyà and colleagues also conducted a cross-sectional study to define health literacy levels of students from Spain and France, highlighting that 36.5% of students had sufficient health literacy [67].

In addition, sense of coherence (SoC) is also considered as a health promotion resource, particularly amongst the youth. A high SoC protects people from stress that supports mental well-being. Chu and colleagues [68] conducted a research with Chinese university students to assess the association and effects of SOC on perceived stress and lifestyle. This study found that higher SOC is positively associated with social support and better performance, however, it is negatively associated with perceived stress. 

Summing up, critical health literacy is mandatory to ensure how individuals integrate their knowledge or information into optimal healthy behaviors. The association of health literacy with increased morbidity and mortality for non-communicable diseases is already evident from previous research [69]. Similarly, the significance of health literacy was equally highlighted for communicable diseases, such as COVID-19 [70]. Further, COVID-19 also requires individuals and collective readiness and adaptive ability to develop critical health literacy, along with prevention and management to ensure compliance. Given the local context of Pakistan, more thorough and comprehensive research is required on the topic to have a better picture of the prevalence of DHL in view of COVID-19 amongst university students, leading to more knowledgeable and informed health decisions.

## 3. Materials and Methods

### 3.1. Study Design and Setting

We conducted an online-based cross-sectional study among university students enrolled in any of the bachelor’s, master’s, MPhil or doctorate programs in Punjab province, Pakistan. Punjab is the largest province with respect to population size and contains around 53% of the total population of Pakistan. There are 30 general public sector universities in Punjab province. Four universities were selected randomly from the list of 30 general public sector universities. The sample size was calculated on the basis of the size of population (700,000), a fraction of 0.5 and a confidence level of 0.95. For the sample of 1979 respondents, the level of sampling error was 2.2%.

The survey was conducted as part of COVID-HL research consortium, a network of researchers on HL from 49 countries, by the Department of Public Health, University of the Punjab and Department of Management Sciences, University of Okara. The questionnaire developed by COVID-HL research consortium was converted to Survey Monkey (an online survey tool). Survey monkey was used as it provides easy access and fast distribution at a low cost. It has the features to prevent input and data coding errors and to provide information on response rates and time duration for completion of the questionnaire. Few modifications were made in the questionnaire to adapt it to the country′s educational structure. The link along with an invitation letter for participation in the survey was shared with the deans and heads of different departments in all four universities. These heads then introduced the study and shared the link for participation in the study with all the students in their respective departments through WhatsApp groups, Facebook and also directly on their emails. Equal chance was given to both male and female students to participate in the study. In Pakistan, the female-to-male student ratio in higher educational institutions is 0.87. The data collection continued until the desired sample size was achieved.

The study was conducted as an online-based survey because of the COVID-19-related lockdown in the country at the time of survey. Pre-testing of the questionnaire was done by sending a link to students in contact with the first author. The data of pre-testing was not included in the final analysis. The study was conducted from 1 May 2020 to 15 June 2020. A reminder message was sent after two weeks of the start of the survey. Each interview took on average 14 min for completion. The total number of interviews conducted was 1980. However, 233 incomplete questionnaires were not included in the final analysis.

### 3.2. Measures

COVID Health Literacy Survey Questionnaire (COVID-HL-Q) was used for data collection [65,71]. The COVID-HL-Q was developed by COVID-HL research consortium administrator Prof. Dr. Kevin Dadaczynski from Fulda University of Applied Sciences, Germany, and Dr. Orkan Okan from the Interdisciplinary Centre for Health Literacy Research at Bielefeld University [65,71]. Content validity of the instrument was done by a panel of experts in HL. The COVID-HL-Q assesses socio-demographic characteristics such as age (in years), gender (female, male, diverse), and country of birth (Pakistan, another country). Information was also collected regarding students′ course of study related variables such as subject group of study (engineering sciences, linguistic and cultural studies, mathematics/natural sciences, medicine/health sciences, law and economics, and social sciences/social work/psychology/education, any other), name of university (University of the Punjab, Okara University, Government College University Faisalabad, and Quaid-e-Azam University), study program (bachelor’s, master’s, MPhil, PhD, any other), current semester of studies, financing of studies (support by parents, by other family member, employment during the semester or during semester break, scholarship, any other), and satisfaction with financial situation (completely sufficient, sufficient, less sufficient, not sufficient). 

Subjective social status of students was measured by using the MacArthur Scale [72]. It is a single-item measure that assesses a person′s perceived social status with respect to others in their group. At the top of the ladder are the people who are the best-off as they have more money and the best job. The students were asked to think of a ladder as representing where people in their country stand and they were asked where they would place themselves on this ladder on a ten-point scale (1–10). 

Sense of coherence (SoC) was measured using the 9-item scale developed by Vogt, Jenny and Bauer [73]. The original scale was developed within the work context; however, it was verbally adapted to assess the current living situations of students. The scale measures the three domains of SoC: comprehensibility (manageable, structured, clear, predictable), manageability (easy to influence, controllable), and meaningfulness (meaningful, significant, rewarding). The responses were taken through a 7-point Likert scale (strongly agree to strongly disagree). Cronbach′s alpha of the overall scale was 0.89; for the subscales it ranged from 0.87 to 0.89. 

Information-seeking behavior related to COVID-19 was measured by asking whether “the students have searched the internet in the last four weeks for information about the coronavirus.” Students′ HL regarding COVID-19 was assessed by using the Digital Health Literacy Instrument (DHLI) of Van der Vaart and Drossaert [34]. It was utilized to assess respondents′ skills in information searching, adding self-generated content, evaluating reliability, determining the relevance of COVID-19 related information and protecting privacy [34]. Overall, the DHLI comprised 15 items with 1–4 response options (very easy–very difficult for items 1–12 and never–often for items 13–15). A lower score represents higher level of DHL of students. Cronbach′s alpha of the scale was 0.88.

The students were also asked about different sources used for online information searching on a 1–4 response scale (never–often), language of the sources used for searching COVID-19-related information (English, Urdu, both Urdu and English) and specific topics used for searching COVID-19-related information. They were also asked about the importance of the searched information and their satisfactions with the information found on the internet regarding COVID-19.

### 3.3. Data Analysis

The data were analyzed using Statistical Package for Social Sciences (SPSS) version 26. Univariate analysis was done, and descriptive statistics were presented as frequencies, percentages, means and standard deviations for socio-demographic characteristics, students′ studies-related variables, and digital-health-related characteristics. The mean score was used as a cut-off point for SoC and DHL. Bivariate analysis was done using ANOVA and simple linear regression to see the association between socio-demographic characteristics and DHL. Finally, a multiple linear regression was employed to assess the association. Only the variables significant at *p* < 0.05 at bivariate analysis were placed in multivariate analysis. A *p*-value <0.05 was considered statistically significant. The Variance Inflation Factor was used to confirm that no multicollinearity between independent variables existed.

### 3.4. Ethical Considerations

The study protocols were reviewed and approved by the Institutional Ethical Review Board at the University of the Punjab (reference No. 132/IERB/PU/2020). Informed written consent was taken before filling out the questionnaires. All respondents were explained about their voluntary participation and assured about data privacy and anonymity. Before the start of questionnaire, students were given information about the study’s objectives and its significance. No personal information such as name, phone number and address were collected to ensure the respondents’ privacy.

## 4. Results

### 4.1. Sample Characteristics

Table 1 shows that a total of *n* = 1747 respondents participated in the survey and completed the self-administered questionnaire. The mean age of students was 22.5 years (SD + 4.5). Overall, 52.7% were females and almost all (99.7%) were born in Pakistan. The majority of respondents were studying the courses of mathematics/natural sciences (32.9%), social sciences (23.7%) and engineering sciences (21.3%), under the bachelor’s (52.4%), master’s (28.1%) and MPhil/PhD (19.5%) programs, within the University of Okara (73.4%) and University of the Punjab (19.2%). Regarding students′ primary financing sources for studies, more than three-quarters of respondents stated that they were supported by their parents (76.8%). Most of the students were found to be satisfied on having sufficient finances (47.0%), however, few reported having non-sufficient funds (10.9%) at their disposal.

This research applied the MacArthur Scale of Subjective Social Status to capture the respondents′ placement on a social ladder between a score of 1 to 10, indicating top score for best-off and bottom for worst-off. The mean of the MacArthur scale was 5.3 (SD + 2.2). The findings revealed that around 21% and 50% of respondents belonged to the subjective social status between 1–3 and 4–6 score in the top ladder for best-off. Figure 1 displays the respondents′ individual scores on the social ladder.

### 4.2. Sense of Coherence

Respondents′ SoC was determined through assessing their aptitude to apply existing and potential resources to cope with COVID-19 related stress and promote health. Table 1 presents the measurement of SoC, based on students′ perception about three broader dimensions, i.e., comprehensibility (cognitive dimension), manageability (behavioral dimension) and meaningfulness (motivational dimension). Findings revealed that the vast majority of students had an overall high level of SoC (59.8%), which is the coping capacity to deal with COVID-19 stress. Similarly, most of the respondents had a high level of comprehensibility (59.6%), manageability (63.5%), and meaningfulness (56.8%) to combat health-related stress. 

Furthermore, Table 2 highlights the item-wise results of the three broader dimensions of SoC through psychometric evaluation among study participants against a 7-point Likert scale (strongly agree to strongly disagree) to infer respondents’ existing life situations. Results show that a significant number of students were found to (strongly/somewhat) agree towards their current life situations and SoC, which are significant (76.1%), meaningful (75.5%), manageable (74.3%), controllable (71.1%), clear (70.0%), rewarding (69.7%) and structured (69.0%), but to a slightly lesser extent easy to influence (67.5%) and predictable (65.4%).

### 4.3. Digital Health Literacy about COVID-19

Respondents′ DHL in relation to COVID-19 was assessed using a number of questions, including purpose, key search areas, sources/means, main topics, frequency and language of searching online information during the four weeks preceding the survey, e.g., number of infected cases and ways to avoid or deal with COVID-19 in daily life. Further, their perception regarding the importance of information and extent of satisfaction was also explored.

Initially, the purpose of searching online information was assessed. Most students reported that they have searched for COVID-19 information on the internet for themselves and other people (55.7%) within four weeks prior to the survey; additionally, 21.4% search for information only for themselves and 7.0% only for other people. Only 15.9% of students did not search for any information related to COVID-19 on the internet during the previous four weeks.

In order to determine participants′ DHL, we used an instrument (DHLI) to measure students′ skills in searching for COVID-19 related information on the internet, under the following five key dimensions: information searching (DHL search), adding self-generated contents (DHL content), evaluating reliability (DHL reliability), determining relevance (DHL relevance) and protecting privacy (DHL privacy) against a 4-point Likert scale (very easy to very difficult). The findings show that the majority of respondents found the online search for information easy or even very easy, particularly in using the proper words/search query (76.2%), making a choice for information to find (73.2%), and finding the exact information they have been looking for (64.0%). Similarly, most of the respondents claimed that they found DHL content (very) easy regarding adding self-generated content about COVID-19 on various forums or social media platforms (e.g., Facebook or Twitter), including posting messages for people to understand exactly what is meant (67.9%), expressing opinion/thoughts/feeling in writing (63.0%), and clearly formulating question (62.7%). Contrary to the above, a large number of students informed about the difficulty to evaluate the reliability of online information (DHL reliability), in terms of deciding whether the information was reliable (64.5%) or whether the information was attached to commercial interests (53.9%), and verifying the information from different websites (55.9%) (Table 3).

Furthermore, the vast majority of students reported DHL relevancy as (very) easy, while deciding about the applicability of the information (68.5%), particularly for healthcare-related decision-making (69.9%) and in daily life (68.8%). Lastly, most of the students declared that they never found DHL privacy difficult, while posting messages on a public forum or social media about COVID-19, specifically in judging who can read the posted message (33.5%), sharing their own (62.5%) or some else′s private information (71.7%) (Table 3).

Upon computing the items described above, the mean score for the overall DHL was 33.1 (SD + 6.1), indicating that on average participants had a high level of DHL to deal with COVID-19-related online-based information. Similarly, the higher mean values for various items of DHL were recorded for DHL reliability (M = 7.6, SD + 2.3), DHL contents (M = 6.8, SD + 1.8), DHL relevance (M = 6.6, SD + 1.7), DHL search (M = 6.5, SD + 2.1) and DHL privacy (M = 5.4, SD + 2.1). Of all students, 54.3% had an overall high DHL. Likewise, the vast majority of respondents exhibited high levels of DHL contents (66.0%), followed by DHL search (57.1%), DHL relevance (55.0%), DHL privacy (53.5%) and DHL reliability (44.4%).

Furthermore, this study explored the frequency of searching for online information from various sources/means about COVID-19 (Table 4). The findings revealed that a significant number of young students often sought information through search engines (e.g., Google, Bing, Yahoo; 43.8%), social media (e.g., Facebook, Instagram, Twitter; 39.9%), YouTube (39.7%) and news portals (e.g., newspapers, TV news channels; 36.7%). Nevertheless, respondents sometimes also searched national websites (e.g., National Command Operations Center (NCOC); 26.5%), Wikipedia and other online encyclopedias (26.1%), and multiple health blogs (29.4%). Additionally, some of the students informed that they had never searched guidebook communities (35.6%), health portals (31.1%) or doctors/pharmaceutical websites (34.6%).

Furthermore, in response to the use of language for seeking online information about COVID-19, the majority of students reported using the English language (63.7%). Nonetheless, few respondents either used both English and Urdu languages (19.3%), or Urdu only (17.0%). 

Upon exploring the specific search topics for COVID-19 (Table 5), the vast majority of respondents stated that they searched for the current spread of disease/infected cases (57.5%) and symptoms of COVID-19 (15.1%). However, only few students explored the information about transmission routes (5.9%), protective measures (5.2%), economic and social consequences of COVID-19 (3.2%) and others (5.2%).

The findings presented in Table 6 divulged that most of the students considered the importance of information according to the fact whether they are up-to-date (68.7%), verified (62.6%) or official (62.4%). A relatively high number of respondents prioritized that information which enabled them to quickly learn significant things (56.2%), which is comprehensive (48.4%) and represents different opinions (42.6%).

Lastly, the respondents′ extent of satisfaction regarding online information about COVID-19 was inquired against a 5-point Likert scale, where the vast majority of students were found either very satisfied (25.6%), satisfied (35.0%), or partially satisfied (25.2%). Nevertheless, few students were dissatisfied (11.2%) or even very dissatisfied (3.0%).

### 4.4. Relationship between Key Characteristics and Digital Health Literacy

Table 7 presents the cross-tabulation to explore the relationship between sociodemographic and further characteristics with overall DHL and its five dimensions, i.e., DHL search, DHL content, DHL reliability, DH relevancy and DHL privacy. Overall, a higher mean score of DHL was found among the students between 17–20 years (M = 33.45, SD + 6.3), females (M = 33.57, SD + 6.0), studying in the University of Okara (M = 33.48, SD + 6.3), studying law and economics (M = 33.77, SD + 5.7), being in a bachelor’s program (M = 33.19, SD + 5.8), suffering from chronic disease (M = 33.53, SD + 6.5) or any disability (M = 33.27, SD + 7.1), belonging to relatively low subjective social class (M = 33.90, SD + 6.6), having a low sense of coherence (M = 34.51, SD + 6.1), and being somewhat satisfied with information (M = 35.08, SD + 5.5). A significant relationship of overall DHL (*p* < 0.05) was observed with gender, studying university, subjective social status, sense of coherence, satisfaction and importance of information (Table 7).

### 4.5. Bivariate and Multivariate Linear Regression of Key Characteristics with Digital Health Literacy

Table 8 illustrates the bivariate and multivariate linear regression of sociodemographics and other characteristics with overall DHL. Simple linear regression was performed to establish the relationship between DHL and key characteristics of respondents. A statistically significant association (*p* < 0.05) was observed with all variables. Furthermore, a multivariate logistic regression analysis was carried out. Results show that students’ gender, sense of coherence, satisfaction and appraisal of the importance of information were found to be significantly associated with overall digital health literacy. Multivariate analysis revealed that female students showed 0.93 increase in DHL (95% CI: 0.28–1.56), where a unit increase in sense of coherence was associated with a 0.13 unit increase in their DHL (95% CI: 0.09–0.16). Similarly, a higher likelihood of 2.02 units of DHL was seen among those respondents, who gave importance to information (95% CI: 1.37–2.68). However, a negative association was observed with students’ satisfaction with information (β = −1.05; 95% CI: −1.49–0.59). The R^2^ of 0.13 indicated a good model fit. 

## 5. Discussion

This study assessed DHL among university students in Pakistan, particularly exploring their information seeking ability and behavior, along with sense of coherence to cope with anxiety. In the time of the COVID-19 pandemic, it is the first kind cross-sectional study representing the young population in Pakistan and analyzing their DHL—and its associated factors—comprehensively.

Among the sociodemographic characteristics, it is worth mentioning that mostly female students participated in this research. This can be accounted for by course or study program type, with relatively highly feminized fields covered under this research. The mean score of MacArthur scale of subjective social status was 5.3, highlighting best-to-moderate positioning of participants within the social ladder. It is because individuals older than 25 years of age are usually employed, thus are more stable in the social ladder [74]. Similar to a Chinese [68] and three Asian countries studies [75], this research reported a high level of sense of coherence amongst students—a psychological measure for health promotion to strengthen coping capacity against COVID-19 stress.

### 5.1. Critical Analysis of Results in the International Context

DHL is a critical and frontline tool to combat the COVID-19 pandemic. Previous research emphasized that people with greater knowledge are more likely to adopt preventive and protective behavior regarding COVID-19 [76,77]. It is argued that health literacy not only empowers individuals and facilitates making informed decisions, but also enhances individual capacities for a collective societal response [61].

Along with traditional sources of information such as friends, family and print media, the internet provides a global platform to seek and understand health information for disease control and prevention. Therefore, this research focused on DHL of university students, who are considered frequent users of the internet. The mean score for overall DHL in our study (33.1) is comparable to Norwegian adolescents, where the mean score of 35.2 was reported for health literacy [75]. Further, this research revealed that more than 50% of the university students perceived themselves to have a higher DHL in relation to COVID-19. These findings are comparable with previous studies conducted in Vietnam [10], Europe [25], Germany [27,78] and Midwestern city [79]. However, they are inconsistent with Pakistan [80], France and Spain [67,81], where students had relatively low to moderate levels of health literacy. Findings reiterate that the participants who had higher DHL may have lower risk of COVID-19 infection. Since this research was carried out during the first wave of the COVID-19 pandemic, it could be argued that government policies, preventive measures and public health interventions were observed strictly during lockdown in Pakistan, similar to other countries [27], which may result in higher DHL. Overall, higher DHL is an indicator of individual wellness and self-efficacy for pandemic-related stress. These findings support the need for continuous awareness raising for the general population and high-risk communities to enhance DHL and endorse public health [79].

The internet facilitates instant access to the latest information from a variety of sources, allowing control over choices and autonomy to the users. However, it also leads to multiple challenges, including low-quality or even unreliable information [82]. The results showed that participants used different means and sources to stay informed during the lockdown. Most of the students searched for information about the spread of infected cases and symptoms for themselves and other people. Interestingly, the majority considered the verified and official information most significant to remain informed and showed satisfaction. These findings corroborate previous similar studies carried out in the context of COVID-19 [79,83,84]. 

The findings also indicate the variance in overall DHL and its five dimensions according to respondents′ sociodemographic- and education-acquisition-related characteristics (e.g., subject groups, programs and primary financing sources). Here, a proxy measure of subjective social status was used as a potential confounder to determine socio-economic status. The results are consistent with previous research, highlighting that sociodemographic and educational factors affect health literacy outcomes [40,67,80]. This research emphasizes the need to address health literacy deficiencies through improving access and providing continuous education, particularly focusing on rural areas. Further, this research also examined that lack of information influences the students′ sense of coherence, which is a protective factor for reducing anxiety. It is evident that the sense of coherence has the potential to manage available resources and promote mental health during stressful situations, particularly amongst students [68].

DHL requires critical operational and navigations skills for searching online information and applying it in daily life. Considering the diversity of health information, DHL was categorized into five broader dimensions (search, content, reliability, relevance and privacy). Our study results show a higher DHL behavior for COVID-19 (~50%) amongst Pakistani students in all dimensions, except for DHL reliability. The dimensions of appraising health information in terms of evaluating reliability and determining relevance are considered more complex competencies, also known as critical HL [40]. Regarding DHL search and content of COVID-19, this research reported that less than one-third of students reported problems to find out the correct information and formulate questions on a relevant topic. However, greatest challenges were observed for assessing the reliability, particularly in judging the information reliability and comparing websites. For DHL relevancy and privacy, some of the students reported difficulties in applying information in daily lives and evaluating the privacy of posted messages. These challenges highlight the interactive character of the internet and social media platforms, which contribute to misinformation, disinformation, and conspiracy theories [85]. Thus, official or public body websites are more expert-driven, thus enhancing subjective trustworthiness and reliance [82].

This research concludes that health information is accessible from several perspectives and sources, ranging from personal social media posts to scientific data and experts′ opinions, including government officials, media spokespersons, researchers and academics to ensure the quality of health of communities. Along with the progression of pandemic, health literacy has become a serious concern due to the influx of information, which may raise problems in findings and applying appropriate information. Although the internet provides the opportunity to verify statistics, deliver high-quality evidence and consult technical experts, inaccurate and misleading information may also result into risky or harmful behavior in the worst cases, regardless.

### 5.2. Limitations

The study has some limitations. Within the cross-sectional design, we only captured findings related to the first wave of the COVID-19 pandemic. Since COVID-19 is a highly demanding field with rapid changes, it covers only the status of DHL at a specific point in time. Because of the online-based assessment, we could not include proportionate samples from each university. One might expect that students who are familiar with the internet due to the online-based study conduction were the ones more likely to take part in the study. This might lead to a bias in results, although the age group in general and students in particular are overall familiar with the internet. Another limitation refers to the fact that all information is self-reported. This limitation is particularly relevant for social status and DHL. In addition, one needs to keep in mind that DHL has not been assessed by using an instrument testing the functional health literacy. However, the study is based on a tool that has been validated in the international context.

### 5.3. Future Directions

This research emphasizes that health literacy is critical and a matter of study today. Hence, it is recommended to apply the same instrument to the general public, preferably amongst older adults or respondents within a higher age range. Since people of older ages are more responsible for healthcare decisions than university students, it would add value in the existing literature and would also provide opportunity to compare the results between varied age groups.

Future research should focus on the relationship between health literacy and actual health behaviors of various age groups. It will be beneficial to design more focused interventions, for instance, designing varied health messages to increase knowledge among people of different health literacy levels as well as to motivate them to take certain health-related decisions for improving quality of life.

## 6. Conclusions

The research provides an insight into university students′ DHL in Pakistan, their approach to seeking health information and behavior related to COVID-19 for promoting both personal and community health. This research also provides insights relevant to the COVID-19 crisis, requiring immediate attention of government, communities and individuals to adopt protective behavior and cope with complex health situations. Firstly, the government should invest more in health education and initiate focused interventions of health promotion and preparedness in response to the prevailing situation. Secondly, structural and interdisciplinary approaches should be adopted to engage communities and media through awareness raising and health education programs to enhance DHL in view of the infodemic. Thirdly, universities should expand digital learning opportunities to improve DHL and reinforce digital skills and competences amongst students of Pakistan.

Since students are the agents of information, this research calls for a program to promote DHL, particularly involving students from rural areas and high-risk communities in Pakistan. It is envisaged that evidence of this research would be beneficial for policy makers, health educators and public health practitioners engaged in health literacy programs for informed decision making, as well as to improve and enhance DHL. Health literacy requires individual and system preparedness to resolve complex real-life issues. Amid the pandemic and infodemic, COVID-19-related health literacy is an underestimated problem—a key indicator to acquire necessary health information and adopt protective behaviors.

## Figures and Tables

**Figure 1 ijerph-18-04009-f001:**
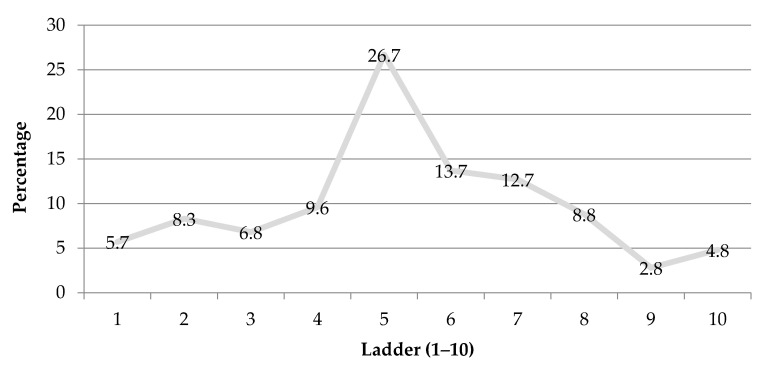
Subjective social status.

**Table 1 ijerph-18-04009-t001:** Characteristics of respondents (*n* = 1747).

Characteristics	*n* ^1^	%
Age of respondents	Mean 22.5	SD + 4.5
17–20 years	652	37.3
21–24 years	789	45.2
>24 years	306	17.5
Gender		
Male	826	47.3
Female	920	52.7
Study subject groups		
Engineering sciences	371	21.3
Mathematics/Natural sciences	574	32.9
Linguistic and cultural studies	51	2.9
Medicine/Health sciences	68	3.9
Law and economics	111	6.4
Social sciences	422	23.7
Others	150	8.7
University		
University of Okara	1280	73.4
University of Punjab	336	19.2
Others	131	7.5
Study program		
Bachelor’s	916	52.4
Master’s	491	28.1
MPhil/PhD	340	19.5
Semester currently studying		
1–2	759	43.4
3–4	589	33.7
>4	399	22.8
Subjective social status		
Low	364	20.9
Medium	871	49.9
High	508	29.1
Primary source of financing		
Support by parents	1278	76.8
Student grant	30	1.8
Employment during the semester	211	12.7
Employment during the semester break	5	0.3
Scholarship	39	2.3
Other	101	6.1
Satisfaction with financial situation		
Completely sufficient	244	14.0
Sufficient	817	47.0
Less sufficient	489	28.1
Not sufficient	189	10.9
Overall sense of coherence (SoC)		
High SoC	942	59.8
Low SoC	632	40.2
Comprehensibility		
High	956	59.6
Low	648	40.4
Manageability		
High	1025	63.5
Low	588	36.5
Meaningfulness		
High	917	56.8
Low	697	43.2

^1^ Values do not always add up to *n* = 1747 due to missing values.

**Table 2 ijerph-18-04009-t002:** Percentage distribution of sense of coherence.

Characteristics	Strongly Agree	Agree	Somewhat Agree	Neither Agree Nor Disagree	Somewhat Disagree	Disagree	Strongly Disagree
Manageable (*n* = 1669)	9.7	36.5	28.1	13.7	5.5	2.8	3.8
Structured (*n* = 1637)	6.6	36.3	26.1	17.2	6.2	2.7	4.9
Easy to influence (*n* = 1630)	6.4	35.0	26.1	15.8	7.5	3.4	5.8
Clear (*n* = 1630)	8.1	40.9	21.0	13.1	7.7	3.3	5.9
Controllable (*n* = 1622)	8.7	39.8	22.6	13.9	6.2	3.0	5.8
Predictable (*n* = 1623)	6.7	33.1	25.6	18.1	6.0	3.5	6.9
Meaningful (*n* = 1627)	12.0	41.4	22.1	12.7	4.4	2.5	5.0
Significant (*n* = 1628)	11.2	44.0	20.9	11.7	5.3	2.5	4.4
Rewarding (*n* = 1626)	10.4	36.7	22.6	15.2	5.2	3.3	6.6

**Table 3 ijerph-18-04009-t003:** Digital health literacy among respondents.

Characteristics	Very Easy	Easy	Difficult	Very Difficult
Information searching				
Make a choice from all the found information (*n* = 1470)	24.0	49.2	17.6	9.2
Use the proper words/search query to find information (*n* = 1452)	18.1	58.1	16.3	7.6
Find the exact information they are looking for (*n* = 1450)	16.1	47.9	26	10
Adding self-generated contents				
Clearly formulate question (*n* = 1451)	12.0	50.7	30.9	6.3
Express opinion, thoughts or feeling in writing (*n* = 1454)	11.8	51.2	31.6	5.4
Write message such that people understand exactly what they mean (*n* = 1449)	14.6	53.3	28.6	3.6
Evaluating reliability				
Decide whether the information is reliable or not (*n* = 1466)	16.3	19.2	41.9	22.6
Decide whether the information is written with commercial interests (*n* = 1460)	26.6	19.5	32.8	21.1
Check different websites to see whether they provide the same information (*n* = 1459)	26.5	17.6	36.9	19.0
Determining relevance				
Decide whether the found information is applicable (*n* = 1447)	13.3	55.2	27.2	4.2
Apply the found information in daily life (*n* = 1449)	11.3	57.5	28.2	3.0
Use found information to make health-related decisions (*n* = 1448)	14.3	55.6	26.2	3.9
**Protecting privacy**	**Never**	**Once**	**Several times**	**Often**
Find it difficult to judge who can read the posted message (*n* = 1458)	33.5	22.3	26.0	18.2
Share private info (*n* = 1460)	62.5	14.3	15.2	8.0
Share some else’s private info (*n* = 1456)	71.7	12.6	10.6	5.1

**Table 4 ijerph-18-04009-t004:** Searching for information related to COVID-19 and related topics on the internet.

Characteristics	Often	Sometimes	Rarely	Never	Don′t Know
Search engines (Google, Bing, Yahoo) (*n* = 1426)	43.8	31.1	13.3	6.1	5.8
Websites of public bodies (NCOC, provincial health departments) (*n* = 1415)	22.2	26.5	21.1	19.4	10.8
Wikipedia and other online encyclopedias (*n* = 1407)	20.8	26.1	21.3	23.2	8.6
Social media (Facebook, Instagram, Twitter) (*n* = 1415)	39.9	28.9	12.4	14.0	4.8
YouTube (*n* = 1414)	39.7	29.9	15.8	11.0	3.5
Blogs on health topics (*n* = 1399)	18.9	29.4	21.4	23.2	7.0
Guidebook communities (*n* = 1411)	12.5	20.4	19.8	35.6	11.7
Health portals (*n* = 1405)	14.2	22.8	20.5	31.1	11.5
Websites of doctors/pharmaceutical companies (*n* = 1404)	13.7	21.9	18.9	34.6	10.9
News portal (e.g., newspapers, TV stations) (*n* = 1405)	36.7	34.2	14.5	10.3	4.3

**Table 5 ijerph-18-04009-t005:** Specific topics searched for in the context of COVID-19 (*n* = 1470).

Characteristics	*n*	%
Current spread of COVID-19 (e.g., number of infected cases)	845	57.5
Transmission routes of COVID-19	87	5.9
Symptoms of COVID-19	222	15.1
Individual measures to protect against infection (e.g., hand-washing tips)	76	5.2
Hygiene regulations (e.g., disinfection and cleaning)	23	1.5
Current situation assessments and recommendations	40	2.7
Restrictions (e.g., exit restrictions, stay-at-home orders)	22	1.5
Economic and social consequences of the COVID-19	48	3.2
Dealing with psychological stress caused by COVID-19	31	2.2
Others	76	5.2

**Table 6 ijerph-18-04009-t006:** Search for information related to COVID-19 and related topics on the internet.

Characteristics	Very Important	Rather Important	Rather Not Important	Not at All Important
The information is up to date (*n* = 1411)	68.7	22.6	4.5	4.3
The information is verified (*n* = 1386)	62.6	25.9	6.3	5.3
Quickly learn the most important things (*n* = 1399)	56.2	31.5	6.5	5.9
The information comes from official sources (*n* = 1398)	62.4	23.8	7.9	5.9
Different opinions are represented (*n* = 1391)	42.6	38.2	11.5	7.6
The subject is dealt with comprehensively (*n* = 1392)	48.4	36.2	8.8	6.6

**Table 7 ijerph-18-04009-t007:** Relationship between respondents’ characteristics and digital health literacy.

Characteristics	Overall DHL (*n* = 1344)		DHL Search (*n* = 1431)		DHL Contents (*n* = 1435)		DHL Reliability (*n* = 1450)		DHL Relevance (*n* = 1429)		DHL Privacy (*n* = 1451)	
	*n*	Mean (SD)	*n*	Mean (SD)	*n*	Mean (SD)	*n*	Mean (SD)	*n*	Mean (SD)	*n*	Mean (SD)
Age of respondents	*p* = 0.12		*p* < 0.001		*p* = 0.60		*p* = 0.26		*p* = 0.12		*p* = 0.68	
17–20 years	484	33.45 (6.3)	511	6.78 (2.1)	521	6.91 (1.8)	527	7.61 (2.2)	522	6.70 (1.7)	532	5.36 (2.0)
21–24 years	619	33.14 (5.9)	664	6.54 (2.1)	655	6.83 (1.7)	664	7.62 (2.3)	651	6.66 (1.7)	661	5.55 (2.1)
>24 years	241	32.42 (6.2)	256	6.11 (2.1)	259	6.60 (1.9)	259	7.89 (2.3)	256	6.44 (1.7)	258	5.45 (2.1)
Gender	*p* = 0.04		*p* = 0.04		*p* = 0.005		*p* = 0.03		*p* = 0.33		*p* = 0.68	
Male	630	32.61 (6.2)	670	6.43 (2.1)	674	6.68 (1.8)	679	7.53 (2.2)	671	6.54 (1.7)	683	5.48 (2.0)
Female	714	33.57 (6.0)	761	6.66 (2.1)	761	6.95 (1.7)	771	7.78 (2.2)	758	6.72 (1.6)	768	5.44 (2.1)
Study subject groups	*p* = 0.4		*p* = 0.003		*p* = 0.21		*p* = 0.09		*p* = 0.72		*p* = 0.32	
Engineering sciences	275	33.46 (6.8)	291	6.82 (2.3)	297	6.84 (2.1)	296	7.41 (2.2)	297	6.72 (1.9)	298	5.72 (2.1)
Mathematics/Natural sciences	431	33.23 (5.9)	464	6.67 (2.1)	456	6.94 (1.7)	477	7.61 (2.3)	462	6.66 (1.6)	478	5.37 (2.0)
Linguistic and cultural studies	33	33.75 (7.2)	37	6.89 (2.2)	37	6.86 (1.7)	39	7.48 (2.0)	36	6.72 (1.9)	37	5.45 (2.4)
Medicine/Health sciences	61	33.13 (6.7)	65	6.21 (2.1)	64	6.71 (1.7)	63	7.93 (2.1)	63	6.63 (1.8)	62	5.58 (2.1)
Law and economics	89	33.77 (5.7)	93	6.6 (1.8)	92	7.05 (1.7)	91	8.15 (2.2)	92	6.79 (1.5)	92	5.34 (2.1)
Social sciences	340	32.47 (5.7)	356	6.18 (2.0)	356	6.62 (1.7)	358	7.82 (2.2)	355	6.51 (1.6)	360	5.36 (2.0)
Other	115	33.12 (5.6)	125	6.56 (2.2)	123	6.82 (1.6)	126	7.59 (2.2)	124	6.59 (1.4)	124	5.51 (2.2)
University	*p* = 0.002		*p* < 0.001		*p* = 0.01		*p* = 0.04		*p* = 0.10		*p* < 0.001	
University of Okara	965	33.48 (6.3)	1039	6.75 (2.2)	1039	6.91 (1.8)	1061	7.58 (2.3)	1036	6.74 (1.7)	1060	5.48 (2.1)
University of the Punjab	273	32.37 (5.6)	284	6.02 (1.8)	288	6.68 (1.7)	283	7.97 (2.2)	285	6.49 (1.5)	283	5.28 (1.9)
Other	106	31.76 (5.4)	108	6.02 (1.8)	108	6.42 (1.9)	106	7.03 (2.0)	108	6.06 (1.6)	108	5.76 (2.1)
Program of studies	*p* = 0.88		*p* = 0.08		*p* = 0.25		*p* = 0.09		*p* = 0.82		*p* = 0.89	
Bachelor’s	705	33.19 (5.8)	744	6.66 (2.1)	745	6.89 (1.7)	750	7.54 (2.2)	745	6.66 (1.7)	757	5.44 (2.0)
Master’s	378	33.10 (6.4)	403	6.49 (2.1)	410	6.78 (1.8)	416	7.73 (2.3)	409	6.60 (1.7)	412	5.50 (2.1)
MPhil/PhD	261	32.97 (6.3)	284	6.34 (2.1)	280	6.70 (1.8)	284	7.87 (2.4)	275	6.62 (1.7)	282	5.46 (2.1)
Semester currently studying	*p* = 0.33		*p* = 0.40		*p* = 0.80		*p* = 0.07		*p* = 0.32		*p* = 0.79	
1–2	587	33.23 (6.5)	623	6.63 (2.2)	627	6.81 (1.8)	635	7.60 (2.2)	624	6.71 (1.7)	634	5.53 (2.1)
3–4	443	33.29 (6.0)	478	6.46 (2.0)	475	6.86 (1.8)	483	7.85 (2.3)	475	6.58 (1.6)	485	5.51 (2.0)
>4	314	32.67 (5.4)	330	6.51 (2.0)	333	6.79 (1.7)	322	7.51 (2.2)	330	6.58 (1.6)	332	5.44 (2.1)
Subjective social status	*p* = 0.01		*p* < 0.001		*p* < 0.001		*p* = 0.07		*p* < 0.001		*p* = 0.14	
Low	250	33.90 (6.6)	272	6.97 (2.3)	275	7.04 (1.8)	279	7.39(2.4)	272	1.84 (1.8)	280	5.5 (2.1)
Medium	667	33.27 (5.7)	719	6.61 (2.0)	711	6.93 (1.7)	722	7.68 (2.1)	711	1.60 (1.6)	723	5.36 (2.0)
High	424	32.48 (6.3)	437	6.21 (2.1)	446	6.53 (1.8)	446	7.78 (2.3)	443	1.67 (1.7)	445	5.61 (2.1)
Primary source of financing	*p* = 0.32		*p* = 0.29		*p* = 0.42		*p* = 0.64		*p* = 0.93		*p* = 0.93	
Support by parents	973	33.35 (6.1)	1042	6.61 (2.1)	1039	6.87 (1.81)	1048	7.67 (2.3)	1038	6.66 (1.7)	1038	6.66 (1.7)
Student grant	24	33.45 (5.6)	25	6.40 (1.6)	26	6.65 (1.7)	26	8.30 (2.5)	26	6.81 (1.7)	26	6.80 (1.7)
Employment during the semester	169	32.50 (5.8)	180	6.31 (1.9)	181	6.70 (1.8)	184	7.72 (2.3)	178	6.60 (1.7)	178	6.60 (1.7)
Employment during the semester break	5	35.20 (5.9)	5	7.40 2.1)	5	8.20 (2.7)	5	7.60 (2.5)	5	6.60 (2.3)	5	6.60 (2.3)
Scholarship	33	31.78 (5.3)	33	6.09 (1.8)	34	6.64 (1.7)	35	7.25 (2.3)	34	6.44 (1.7)	34	6.44 (1.7)
Other	75	32.82 (7.3)	79	6.37 (2.3)	82	6.84 (1.7)	82	7.52 (2.2)	79	6.54 (1.9)	79	6.54 (1.9)
Sense of coherence	*p* < 0.001		*p* < 0.001		*p* < 0.001		*p* = 0.22		*p* < 0.001		*p* < 0.001	
High	773	32.06 (5.8)	811	6.11 (1.9)	818	6.46 (1.7)	823	7.71 (2.3)	814	6.29 (1.6)	814	6.29 (1.6)
Low	508	34.51 (6.1)	548	7.11 (2.2)	537	7.31 (1.8)	546	7.56 (2.2)	537	7.12 (1.7)	537	7.12 (1.7)
Chronic disease	*p* = 0.38		*p* = 0.18		*p* = 0.42		*p* < 0.001		*p* = 0.72		*p* = 0.72	
Yes	196	33.53 (6.5)	206	6.73 (2.4)	206	6.92 (1.9)	205	7.19 (2.2)	203	6.60 (1.8)	203	6.60 (1.8)
No	1095	33.11 (6.0)	1153	6.52 (2.1)	1158	6.81 (1.7)	1177	7.78 (2.3)	1159	6.65 (1.7)	1159	6.65 (1.7)
Any disability	*p* = 0.65		*p* = 0.70		*p* = 0.80		*p* < 0.001		*p* < 0.001		*p* = 0.22	
Yes	213	33.27 (7.1)	224	6.50 (2.3)	226	6.84 (2.0)	223	7.20 (2.3)	224	5.87 (2.2)	222	6.75 (1.9)
No	1025	33.06 (5.8)	1078	6.54 (2.1)	1077	6.81 (1.7)	1098	7.74 (2.3)	1103	5.35 (2.1)	1080	6.60 (1.6)
Satisfaction with information	*p* < 0.001		*p* < 0.001		*p* < 0.001		*p* < 0.001		*p* < 0.001		*p* = 0.20	
Dissatisfied	171	34.12 (6.8)	182	7.07 (2.4)	186	7.09 (1.9)	189	7.31 (2.3)	182	7.11 (1.9)	189	5.71 (2.2)
Somewhat Satisfied	316	35.08 (5.5)	342	7.11 (2.0)	335	7.20 (1.7)	346	8.11 (2.1)	337	7.11 (1.6)	346	5.40 (2.0)
Satisfied	806	32.22 (5.9)	838	6.21 (2.0)	847	6.63 (1.8)	851	7.61 (2.3)	847	6.35 (1.6)	857	5.43 (2.1)
Importance of information	*p* < 0.001		*p* < 0.001		*p* < 0.001		*p* < 0.001		*p* < 0.001		*p* < 0.001	
Very important	714	32.08 (5.9)	739	6.11 (2.0)	745	6.55 (1.8)	749	7.83 (2.3)	741	6.36 (1.7)	751	5.27 (1.9)
Less important	531	34.69 (5.8)	563	7.10 (2.11)	558	7.22 (1.7)	570	7.59 (2.2)	560	7.01 (1.6)	575	7.59 (2.2)

**Table 8 ijerph-18-04009-t008:** Bivariate and multivariate linear regression of sociodemographic and further characteristics with overall digital health literacy (*n* = 1344).

Characteristics	Bivariate Linear Regression			Multivariate Linear Regression		
	β (SE)	95% CI	*p*	β (SE)	95% CI	*p*
Age of respondents	−0.08 (0.04)	−0.15–0.01	0.02	−0.02 (0.04)	−0.09–0.05	0.63
Gender	0.96 (0.33)	0.31–1.62	0.004	0.93 (0.33)	0.28–1.56	0.005
University	−0.94 (0.26)	−1.47–0.42	<0.001	−0.48 (0.27)	−1.01–0.05	0.07
Subjective social status	−0.26 (0.07)	−0.41–0.12	<0.001	−0.09 (0.08)	−0.24–0.06	0.27
Sense of coherence	0.16 (0.16)	0.13–0.19	<0.001	0.13 (0.02)	0.09–0.16	<0.001
Satisfaction with information	−1.4 (0.23)	−1.85–0.94	<0.001	−1.05 (0.23)	−1.49–0.59	<0.001
Importance of information	2.60 (0.34)	1.94–3.26	<0.001	2.02 (0.33)	1.37–2.68	<0.001
Age of respondents	−0.08 (0.04)	−0.15–0.01	0.02	−0.02 (0.04)	−0.09–0.05	0.63

## Data Availability

The data presented in this study are available upon reasonable request from the corresponding author.

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
