# Peer review of "COVID-19 and Health Information Seeking Behavior: Digital Health Literacy Survey amongst University Students in Pakistan"

_ijerph, 2021, doi:10.3390/ijerph18084009_

Round 1

Reviewer 1 Report

Thank you for giving me the possibility to review this article. I hope the authors find my comments productive and that they will help them to improve their research work.
In this paper the authors conduct a study through an online-based cross-sectional survey to assess the digital health literacy (DHL) of Pakistani students in the midst of the COVID-19 pandemic.
The keywords are in accordance with the terms used in the research.
Introduction
The introduction section correctly puts the research topic in context and the wording is appropriate and meaningful.
Authors are advised to indicate in line 37 the reference date of the data provided (560,000 confirmed cases).
The penultimate paragraph should explain the objective of the research and, as is the case, comment that a research gap has been detected with respect to the topic being addressed and that is why this study has been carried out. The research objective should be clear and well defined.
Authors are asked to explain in the last paragraph of the Introduction the structure that the article will follow, listing each of the sections it contains.
Literature Review
It is necessary to include in the paper a section that refers to the Literature Review and where the main studies and research related to the research topic that have led the authors to carry out their research are included. It is also necessary to include in this same section the hypothesis of the research, i.e., the problem or gap that has been detected and about which
we have made a prediction that later, in the methodology, we have been able to corroborate or not.
Materials and methods
In this section, the authors explain the sample and the questionnaire used. However, the selection of the sample is not controlled in terms of male and female representation, as it is through a link and at the discretion of the departments of the 4 universities chosen, on which the participation survey is sent. Through this action, the authors lose control of the sample and
of possible biases derived from sending the survey to specific students or profiles.
The measurement variables of the survey are correct to carry out the analysis and the methodologies employed as well as the tools used in the research are explained.
The authors are asked to explain in line 179.
Results
The results are well explained, detailed and presented in a clear and understandable way.
Furthermore, the authors have not limited themselves to transcribing the results contained in the tables but have made brief summaries of the main results within this section.

Correct Covid-19 on line 325 (CVOID-19 is written CVOID-19).
Discussion
In this section the authors correctly make a comparison of the results obtained with their research based on previous studies.
The authors are well aware of the limitations of the research and the reasons for some of them. However, it is recommended that the authors propose new lines of research based on the results obtained.
Conclusion
The information contained in the Conclusion is correct but the authors are asked to explain it further and to reflect the main results obtained with the research. Conclusions should clarify the objectives of the research and highlight whether the research has achieved what was intended by the initial objectives. In addition, the authors should include Pakistan in the first lines of the Conclusion and make clear the scope of the study from the beginning of the section.
References
The references are correct and most of them are current (from 2016 to the present).
It is recommended that the authors include studies such as that of: Van den Broucke, Stephan; Levin-Zamir, D; Schaeffer, D; Pettersen, Sverre; Guttersrud, Øystein; Finbraaten, Hanne; Arriaga, M.; Vrdelja, Mitja; Link, T; Pelikan, Jürgen. (2020). Digital health literacy in general populations – An international comparison. European Journal of Public Health. 30.
10.1093/eurpub/ckaa165.124. where Digital health literacy is addressed to address this issue in the introduction section.
Or others such as: Saura, J. R., Reyes-Menendez, A., & Bennett, D. R. (2019). How to extract meaningful insights from UGC: a knowledge-based method applied to education. Applied Sciences, 9(21), 4603. where new data analysis and visualisation techniques can provide a wealth of information about user behaviour and their use of new technologies.
Also authors such as: Saura, J. R., Debasa, F.; Reyes-Menendez, A. (2019). Does user generated content characterize millennials’ generation behavior? Discussing the relation between sns and open innovation. Journal of Open Innovation: Technology, Market, and Complexity, 5(4), 96. that identify the key factors that characterise the millennial generation,
can be references to be taken into account by the authors of this research to be included in the research regarding the part of the users behaviour in the search for information.

Author Response

Reviewer’s comment 1

“In this paper the authors conduct a study through an online-based cross-sectional survey to assess the digital health literacy (DHL) of Pakistani students in the midst of the COVID-19 pandemic.

The keywords are in accordance with the terms used in the research.”

Authors’ reply

Thank you very much for your appreciation.

Reviewer’s comment 2

“Introduction

The introduction section correctly puts the research topic in context and the wording is appropriate and meaningful.”

Authors’ reply

Thank you very much for appreciation.

Reviewer’s comment 3

Authors are advised to indicate in line 37 the reference date of the data provided (560,000 confirmed cases).

Authors’ reply

The reference date has been provided in the manuscript (please see line 39, page 1).

Reviewer’s comment 4

“The penultimate paragraph should explain the objective of the research and, as is the case, comment that a research gap has been detected with respect to the topic being addressed and that is why this study has been carried out. The research objective should be clear and well defined.

Authors are asked to explain in the last paragraph of the Introduction the structure that the article will follow, listing each of the sections it contains.”

Authors’ reply

The suggested changes have been made in the manuscript in the introduction section (please see lines 110-130, page 3).

Reviewer’s comment 5

“Literature Review

It is necessary to include in the paper a section that refers to the Literature Review and where the main studies and research related to the research topic that have led the authors to carry out their research are included. It is also necessary to include in this same section the hypothesis of the research, i.e., the problem or gap that has been detected and about which we have made a prediction that later, in the methodology, we have been able to corroborate or not.”

Authors’ reply

In response to your comment, we have added a section on literature review in the manuscript where the studies on the topic have been presented in detail (please see lines 131-249).

Reviewer’s comment 6

“Materials and methods

In this section, the authors explain the sample and the questionnaire used. However, the selection of the sample is not controlled in terms of male and female representation, as it is through a link and at the discretion of the departments of the 4 universities chosen, on which the participation survey is sent. Through this action, the authors lose control of the sample and of possible biases derived from sending the survey to specific students or profiles.”

Authors’ reply

The universities were selected randomly from the list of public sector universities in Punjab. In Pakistan, the female to male ratio in higher educational institutions is 0.87. Equal opportunities were provided for both male and female students to participate in the study, as the link for participation in the study was shared with all the students of the respective department through their heads. It is also explained in the text at lines 272-277 on Page 6 in the manuscript. 

Reviewer’s comment 7

“The measurement variables of the survey are correct to carry out the analysis and the methodologies employed as well as the tools used in the research are explained.

The authors are asked to explain in line 179.”

Authors reply

Thank you very much for your appreciation. The line 179 in the previous manuscript is rephrased and is presented at lines 316-319 in the revised manuscript. 

Reviewer’s comment 8

“Results

The results are well explained, detailed and presented in a clear and understandable way.

Furthermore, the authors have not limited themselves to transcribing the results contained in the tables but have made brief summaries of the main results within this section.

Correct Covid-19 on line 325 (CVOID-19 is written CVOID-19).”

Authors reply

Thank you for highlighting this mistake. We had made the correction in the manuscript.

Reviewer’s comment 9

“Discussion

In this section the authors correctly make a comparison of the results obtained with their research based on previous studies.

The authors are well aware of the limitations of the research and the reasons for some of them. However, it is recommended that the authors propose new lines of research based on the results obtained.”

Authors reply

We have added a paragraph on future research directions in the manuscript (please see lines 614-624 at page 18 in the manuscript).

Reviewer’s comment 10

“Conclusion

The information contained in the Conclusion is correct but the authors are asked to explain it further and to reflect the main results obtained with the research. Conclusions should clarify the objectives of the research and highlight whether the research has achieved what was intended by the initial objectives. In addition, the authors should include Pakistan in the first lines of the Conclusion and make clear the scope of the study from the beginning of the section.”

Authors reply

Corrections have been made in the text accordingly.

Reviewer’s comment 11

“References

The references are correct and most of them are current (from 2016 to the present).

It is recommended that the authors include studies such as that of: Van den Broucke, Stephan; Levin-Zamir, D; Schaeffer, D; Pettersen, Sverre; Guttersrud, Øystein; Finbraaten, Hanne; Arriaga, M.; Vrdelja, Mitja; Link, T; Pelikan, Jürgen. (2020). Digital health literacy in general populations – An international comparison. European Journal of Public Health. 30. 10.1093/eurpub/ckaa165.124 where Digital health literacy is addressed to address this issue in the introduction section.

Or others such as: Saura, J. R., Reyes-Menendez, A., & Bennett, D. R. (2019). How to extract meaningful insights from UGC: a knowledge-based method applied to education. Applied Sciences, 9(21), 4603. where new data analysis and visualisation techniques can provide a wealth of information about user behaviour and their use of new technologies.

Also authors such as: Saura, J. R., Debasa, F.; Reyes-Menendez, A. (2019). Does user generated content characterize millennials’ generation behavior? Discussing the relation between sns and open innovation. Journal of Open Innovation: Technology, Market, and Complexity, 5(4), 96. that identify the key factors that characterise the millennial generation, can be references to be taken into account by the authors of this research to be included in the research regarding the part of the users behaviour in the search for information.”

Authors reply

Thank you very much for highlighting the relevant literature. In addition to literature on the subject, all these studies have been cited and referenced in the manuscript. 

Reviewer 2 Report

The proposal presents the results of a study to identify digital health literacy (DHL) related to COVID-19 in Pakistani students. The determination of the DHL was made through a data collection instrument that was applied online through a digital platform. Likewise, the sample size, the characteristics of the students from the 4 selected universities and the statistical tools used (simple linear regression, multiple linear regression, P value, Cronbach's alpha, among others) are presented for the development of the study. In addition, the methodology used and the main results obtained are presented. Finally, a discussion of the identified findings is presented, the limitations of the work are indicated and the conclusions are described. The proposal is well structured, well founded and presents a novel and interesting topic, however, the following aspects were identified:

  1. The introduction indicates the importance and concern of misinformation about COVID-19, which particularly generates in young people an increase in stress and anxiety that can generate other diseases. However, the effects of stress and anxiety particularly on the health of young people are not indicated or highlighted. In addition, are not indicated that others diseases affect the health by the confinement, as well as their main effects.
  2. There is confusion or incorrect style when making the citation [23] since in the same idea the APA style citation is also used, which is incorrect according to the indications of the citation style to be used.
  3. A reduced revision of the state of the art is perceived since only 6 works are presented. It is suggested as far as possible to expand this review and clearly indicate the similarities and mainly the differences of the works found with those of the proposal presented, as well as to carry out a general analysis of the aspects identified in the review of the state of the art.
  4. In relation to the previous point, it is suggested to create a section exclusively to present the review of the state of the art or if is decided to present it also in the introduction, it is suggested to include a paragraph that connects the ideas presented before line 108 with the review of the state of the art. In addition, it is suggested to include the description of the works presented in [34] to [38], [17], [40] to [45] and [49].
  5. Why was the study conducted on young students and not the general public? Why was that age group chosen? These aspects are not clearly indicated in the proposal, it is suggested to highlight the justification more.
  6. Additionally, the criteria used for the selection of the four universities and for the choice of the digital platform used (SurveyMonkey) are not indicated. Why was another digital platform such as Google Forms or another not used?.
  7. On the other hand, it is indicated that the reliability of the instrument was made through Cronbach's alpha which was greater (0.89) than the minimum acceptable value (0.7), but it is not indicated how the validation of the instrument was carried out, this because reliability and validity are related and important aspects in the application of a data collection instrument. What kind of validation did you perform? content, criterion or construct. For example, if it was content, what method was used? expert judgment, Factor Analysis, among others. Was a pilot test carried out? regularly this is done within the validation.
  8. The proposal indicates that a high level of DHL in terms of COVID-19 generates greater awareness in self-care and consequently could increase awareness of family and community protection by having greater knowledge of the care and treatments related to the disease. But how can that claim be guaranteed? Is a high DHL rate in a college student really enough to encourage self-care? How does a high impact on DHL better manage the stress and anxiety generated by confinement?.
  9. Finally, it is suggested to add a paragraph at the end of the conclusions describing the future work to be carried out once this study is finished, that is, what is missing or could be done with the results obtained, perhaps applying the same instrument to the general public with a higher age range since mature people are more responsible than university students and then to make a comparison of the results.

Author Response

The proposal presents the results of a study to identify digital health literacy (DHL) related to COVID-19 in Pakistani students. The determination of the DHL was made through a data collection instrument that was applied online through a digital platform. Likewise, the sample size, the characteristics of the students from the 4 selected universities and the statistical tools used (simple linear regression, multiple linear regression, P value, Cronbach's alpha, among others) are presented for the development of the study. In addition, the methodology used and the main results obtained are presented. Finally, a discussion of the identified findings is presented, the limitations of the work are indicated and the conclusions are described. The proposal is well structured, well founded and presents a novel and interesting topic, however, the following aspects were identified:”

Authors’ reply

Thank you very much for your appreciation.

Reviewer’s comment 1

“The introduction indicates the importance and concern of misinformation about COVID-19, which particularly generates in young people an increase in stress and anxiety that can generate other diseases. However, the effects of stress and anxiety particularly on the health of young people are not indicated or highlighted. In addition, are not indicated that others diseases affect the health by the confinement, as well as their main effects.”

Authors’ reply

As the focus of this study was to assess information seeking behaviors among university students, along with their ability to find relevant information and deal with DHL, and the factors associated with DHL in Punjab, Pakistan, we did not present stress and anxiety dimensions in detail here. In another paper the focus is on misinformation and stress among students. Nevertheless, we have added some information in the manuscript (please see lines 67-68 at page 2).

Reviewer’s comment 2

“There is confusion or incorrect style when making the citation [23] since in the same idea the APA style citation is also used, which is incorrect according to the indications of the citation style to be used.”

Authors’ reply

We have made the corrections throughout in the manuscript.

Reviewer’s comment 3

“A reduced revision of the state of the art is perceived since only 6 works are presented. It is suggested as far as possible to expand this review and clearly indicate the similarities and mainly the differences of the works found with those of the proposal presented, as well as to carry out a general analysis of the aspects identified in the review of the state of the art.”

Authors’ reply

In response to your comment, we have added a section on literature review in the manuscript where the studies on the topic have been presented in detail (please see lines 131-249).

Reviewer’s comment 4

“In relation to the previous point, it is suggested to create a section exclusively to present the review of the state of the art or if is decided to present it also in the introduction, it is suggested to include a paragraph that connects the ideas presented before line 108 with the review of the state of the art. In addition, it is suggested to include the description of the works presented in [34] to [38], [17], [40] to [45] and [49].”

Authors’ reply

In response to your comment, we have added a section on literature review in the manuscript where the studies on the topic have been presented in detail (please see lines 131-249).

Reviewer’s comment 5

“Why was the study conducted on young students and not the general public? Why was that age group chosen? These aspects are not clearly indicated in the proposal; it is suggested to highlight the justification more.”

Authors’ reply

We have included information about the choice of the sample in the introduction and literature review within the manuscript.

Reviewer’s comment 6

“Additionally, the criteria used for the selection of the four universities and for the choice of the digital platform used (SurveyMonkey) are not indicated. Why was another digital platform such as Google Forms or another not used?”

Authors’ reply

Four universities were selected randomly out of the list of public sector general universities. The reasons for the selection of survey monkey has been added in the manuscript (please see lines 266-269 at page 6).

Reviewer’s comment 7

“On the other hand, it is indicated that the reliability of the instrument was made through Cronbach's alpha which was greater (0.89) than the minimum acceptable value (0.7), but it is not indicated how the validation of the instrument was carried out, this because reliability and validity are related and important aspects in the application of a data collection instrument. What kind of validation did you perform? content, criterion or construct. For example, if it was content, what method was used? expert judgment, Factor Analysis, among others. Was a pilot test carried out? regularly this is done within the validation.”

Authors’ reply

Yes, pretesting of the questionnaire was done before the start of data collection (lines 279-281). The COVID-HL-Q was developed by the COVID-HL research consortium, a research network comprising 49 countries. COVID-HL-Q was used in different countries like Germany, USA, Poland and others to explore the same questions. Content validity of the instrument was done by a panel of experts in HL.

Reviewer’s comment 8

“The proposal indicates that a high level of DHL in terms of COVID-19 generates greater awareness in self-care and consequently could increase awareness of family and community protection by having greater knowledge of the care and treatments related to the disease. But how can that claim be guaranteed? Is a high DHL rate in a college student really enough to encourage self-care? How does a high impact on DHL better manage the stress and anxiety generated by confinement?”

Authors’ reply

As the focus of this study was to assess information seeking behaviors among university students, along with their ability to find relevant information and deal with DHL, and the factors associated with DHL in Punjab, Pakistan. For that reason, we did not present stress and anxiety dimensions in detail here.

Reviewer’s comment 9

“Finally, it is suggested to add a paragraph at the end of the conclusions describing the future work to be carried out once this study is finished, that is, what is missing or could be done with the results obtained, perhaps applying the same instrument to the general public with a higher age range since mature people are more responsible than university students and then to make a comparison of the results.”

Authors’ reply

We have added a paragraph on future research directions in the manuscript (please see lines 614-624 at page 18 in the manuscript).

Round 2

Reviewer 1 Report

The authors have improved the research following suggestions 

Reviewer 2 Report

In general terms, all observations were attended by authors. The enlarge and reorganization of the manuscript is more appropiate because the main contribution is more clear for the readers of the journal. I have only a minor comment, please be homogeneous when using the term sense of coherence (SoC), lines 235 - 240.